# Nanotechnology and Nanocarrier-Based Drug Delivery as the Potential Therapeutic Strategy for Glioblastoma Multiforme: An Update

**DOI:** 10.3390/cancers13020195

**Published:** 2021-01-07

**Authors:** Jen-Fu Hsu, Shih-Ming Chu, Chen-Chu Liao, Chao-Jan Wang, Yi-Shan Wang, Mei-Yin Lai, Hsiao-Chin Wang, Hsuan-Rong Huang, Ming-Horng Tsai

**Affiliations:** 1College of Medicine, Chang Gung University, Taoyuan 333, Taiwan; jeff0724@gmail.com (J.-F.H.); kz6479@cgmh.org.tw (S.-M.C.); bluewing0356@msn.com (C.-C.L.); cjwang@cgmh.org.tw (C.-J.W.) B9302003@cgmh.org.tw (Y.-S.W.); lmi818@msn.com (M.-Y.L.); cyndi0805@yahoo.com.tw (H.-C.W.); qbonbon@gmail.com (H.-R.H.); 2Division of Pediatric Neonatology, Department of Pediatrics, Chang Gung Memorial Hospital, Taoyuan 333, Taiwan; 3Department of Medical Imaging and Intervention, Chang Gung Memorial Hospital, Taoyuan 333, Taiwan; 4Division of Pediatric Hematology/Oncology and Neonatology, Department of Pediatrics, Chang Gung Memorial Hospital, Yunlin 638, Taiwan

**Keywords:** glioblastoma multiforme, liposomal doxorubicin, target therapy, nanoparticle, immunotherapy

## Abstract

**Simple Summary:**

Glioblastoma multiforme (GBM) are among the most lethal tumors. The highly invasive nature and presence of GBM stem cells, as well as the blood brain barrier (BBB) which limits chemotherapeutic drugs from entering the tumor mass, account for the high chance of treatment failure. Recent developments have found that nanoparticles can be conjugated to liposomes, dendrimers, metal irons, or polymeric micelles, which enhance the drug-loaded compounds to efficiently penetrate the BBB, thus offering new possibilities for overcoming GBM stem cell-mediated resistance to chemotherapy and radiation therapy. In addition, there have been new emerging strategies that use nanocarriers for successful GBM treatment in animal models. This review highlights the recent development of nanotechnology and nanocarrier-based drug delivery for treatment of GBMs, which may be a promising therapeutic strategy for this tumor entity.

**Abstract:**

Glioblastoma multiforme (GBM) is the most common and malignant brain tumor with poor prognosis. The heterogeneous and aggressive nature of GBMs increases the difficulty of current standard treatment. The presence of GBM stem cells and the blood brain barrier (BBB) further contribute to the most important compromise of chemotherapy and radiation therapy. Current suggestions to optimize GBM patients’ outcomes favor controlled targeted delivery of chemotherapeutic agents to GBM cells through the BBB using nanoparticles and monoclonal antibodies. Nanotechnology and nanocarrier-based drug delivery have recently gained attention due to the characteristics of biosafety, sustained drug release, increased solubility, and enhanced drug bioactivity and BBB penetrability. In this review, we focused on recently developed nanoparticles and emerging strategies using nanocarriers for the treatment of GBMs. Current studies using nanoparticles or nanocarrier-based drug delivery system for treatment of GBMs in clinical trials, as well as the advantages and limitations, were also reviewed.

## 1. Introduction

### 1.1. Clinical Features of Glioblastoma Multiforme (GBM)

Glioblastoma multiforme (GBM) is the most common malignant brain tumor, accounting for approximately 50% of all primary malignant tumors in the central nervous system [1,2]. GBM consists of de novo (primary) GBM, arising from normal glial cells, and secondary GBM, developing from an existing low-grade diffuse or anaplastic astrocytoma [2,3,4]. The median survival after diagnosis of GBM is 12.5–18 months and the five-year survival is reported to be approximately 4–7%, despite the launch of new advanced targeted therapy and immunotherapy [5,6,7,8]. The median age at diagnosis of GBM is mid 60 s; males have a 1.6 times higher incidence than females. The incidence rate of GBM is 1.1–5.0 cases per 100,000 per year [7,8,9,10] and is highest among Caucasians compared to that among Africans and Asians [11,12]. However, recent reports found a sustained and statistically significant increase of GBM across all ages [9,10]. Clinical manifestations of GBM are initially nonspecific, including headaches, nausea, memory loss, personality changes, and seizures. Patients may progress rapidly to unconsciousness when the tumor grows to a very large size [7].

Magnetic resonance imaging (MRI) is the first-line imaging examination procedure for diagnosing GBM. It provides information on a tumor’s location, boundary, size, and characteristics. GBM imaging features are often present as ring-enhancing mass lesions, and are characterized by hypointensity on T1-weighted images and heterogeneous enhancement following contrast infusion (Figure 1) [13]. The standard T2-weighted (T2w), T2-fluid-attenuated inversion recovery (T2-FLAIR) (Figure 2), and T1-weighted contrast-enhanced sequences are typically arranged, showing the important characteristics of the mass, including vascularity, single foci, multiple foci, the necrotic region, and changes in brain structure due to tumor compression [14,15]. The typical features of GBM can be observed in other disease entities, such as metastasis, malignant lymphoma, or infective abscess [16]. For better delineation of tumors and disease monitoring of tumor progression, 18F-fluorodeoxyglucose (FDG) positron emission tomography (PET) is currently the most common imaging diagnostic tracer for GBM diagnosis and assessment of early therapeutic responses [13,17,18]. Patients with increased FDG uptake are found to be associated with a poor survival rate [17]. Because GBMs show variously high intratumor heterogeneity, especially in primary GBM [19,20], and it is challenging to obtain multiple regions of a tumor for analyses, the 3D quantitative image features of GBMs after radiogenomic analysis have been developed to identify GBM phenotypic subtypes and predict patient prognosis [17,18,21,22].

In gross patterns, GBMs demonstrate a diversity of morphological features. Most GBMs have extensive vascularity, high endothelial proliferation, high cell density mixed with various necrosis, and some atypia [23,24]. The 2016 World Health Organization (WHO) classification of CNS tumors uses molecular parameters and histology to define brain tumors, which formulate the concept that the diagnosis of CNS tumors should be structured in the molecular era [25]. According to the 2016 WHO classification, GBM can be divided into isocitrate dehydrogenase (IDH)-wild type (90% of all cases), IDH-mutant (10%), and IDH not otherwise specified (in cases without diagnostic procedure or those IDH cannot be performed). The IDH-wide type GBM corresponds to primary or de novo glioblastoma and commonly occurs in adults over 55 years old, while the IDH-mutant GBM corresponds to secondary GBM arising from preexisting astrocytoma and is predominantly found in younger patients [25,26].

### 1.2. Current Treatment

GBM is one of the most lethal cancers with a very poor 5-year survival rate despite advanced therapeutic options in chemotherapy, immunotherapy, and radiation [27]. The current standard policy of GBM includes maximal surgical resection if possible, followed by a combination of chemotherapy and/or radiotherapy. Maximal surgical resection is the first step after confirmation through medical imaging. Radiographic total resection is the most prognostic and a higher extent of tumor resection (>90%) is significantly associated with better one-year survival [28,29]. In the current temozolomide (TMZ) era, maximal resection also confers a significant overall survival benefit in patients with recurrent and resectable GBM [29].

Radiotherapy has been used in patients with residual tumors for more than several decades [30]. Radiation therapy causes tumor cell apoptosis through DNA double-strand breaks [31]. However, approximately 50% of GBMs express amplification of the epidermal growth factor receptor (EGFR) gene, whose truncated variant III, EGFRvIII, which is expressed in nearly one-fourth of all GBMs, confers resistance of GBM to radiation by promoting the rapid repair of DNA double-strand breaks [32]. Furthermore, radiotherapy of brain tumors could promote tumor recurrence or trigger secondary gliomas [33,34]. A recent study demonstrated that radiation-induced DNA double-strand breaks combined with preexisting tumor suppressor losses could contribute to the development of high-grade gliomas in both in vivo and in vitro models [33]. Currently, it is potentially possible to significantly improve GBM therapy by combining ionizing radiation and bioactive DNA repair inhibitors [35]. While radiosurgery techniques have been developed in recent years, stereotactic radiosurgery can confine treatment to the targeted tumor site. Therefore, the role of stereotactic radiosurgery in recurrent GBM has been documented to be significantly associated with longer overall survival and/or progression-free survival [36].

The current standard chemotherapy for GBM is TMZ. Radiotherapy plus concomitant and adjuvant TMZ for GBM was first described in 2005 [37]. It was found to provide better survival outcomes than radiotherapy alone among patients who underwent surgical resection [37,38] and those who only received biopsy [39]. TMZ is an alkylating agent that induces tumor cell apoptosis by methylating the purines of DNA. The ineffectiveness of TMZ comes from O^6^-methylguanine-DNA methyltransferase (MGMT) expression. MGMT is a DNA repair protein that can reverse the TMZ-induced alkylation process and has emerged as a predictor of responsiveness to alkylating agents [40]. Furthermore, TMZ-induced DNA damage in healthy cells causes significant concern. Given the presences of disadvantages and concerns arising from the current standard care of GBM, namely concomitant chemotherapy after maximal resection, novel therapeutic options are urgently needed to improve the treatment efficacy and target the GBM tumor cells.

## 2. Obstacles of GBM Treatment and the Resolution

### 2.1. GBM Stem Cells

The most common mechanism of GBM resistances is the presence of stem-like glioblastoma stem cells (GSCs) and poor permeability restricted by the blood brain barrier (BBB) for most chemotherapeutic agents. GSCs are functionally defined and distinguished from their differentiated glioblastoma cell progeny by the properties of tumor-initiating capacity following serial transplantation, self-renewal, and the ability to recapitulate tumor heterogeneity [41]. The origin of GSCs remains controversial, but it is believed that these progenitor cells arise from neural stem cells or are transformed astrocytes that gain access to stem-specific transcriptional programs [41,42]. The majority of therapeutic modalities to target GSCs have failed during clinical trials, because GSCs have various epigenetic and posttranscriptional regulations that can drive GSCs differentiation, invasive growth, and support GSC maintenance [41,42,43]. GSCs also have high metabolic power to support the rapid proliferation and adapt to harsh microenvironments [44].

The heterogeneity of GBMs further increases the difficulty of treatment. Recent advances in sequencing techniques found the complete genomic landscape of GBMs and revealed profound heterogeneity of individual tumors even at the single cell level [45]. For example, the EGFR genes have been found to have amplifications and mutations in more than half of GMBs, which frequently result in the ineffectiveness of anti-EGFR therapies [46]. Current researchers have tried to find specific biomarkers for GSC populations to distinguish them from non-GSC population in order to target GSCs and sensitize tumors to conventional treatment [47,48]. Cell membrane surface antigens are ideal biomarkers to which antitumor agents can easily bind, leading to increased therapeutic efficacy [48]. However, the optimal markers for GSCs have not yet been identified. Potential biomarkers for GSCs include CD133, CD15/SSEA-1, CD44, integrin-α, and A2B5. Some of these biomarkers can also be used as an indicator of therapeutic response and a prognostic index of GBMs [49].

### 2.2. Transport across the Blood-Brain Barrier

Another obstacle comes from the low permeability of the BBB which makes the delivery of drugs to the intracranial tumors very difficult [50]. The presence of tight junction complexes in the BBB, which lines the endothelial cells of brain capillaries, results in the absence of pinocytosis and fenestrations and reduces permeability to anticancer agents [51]. Furthermore, active efflux transporters (AETs) will vehicle drugs back to the blood and the presence of metabolizing enzymes further makes drugs inactive before they can be released to the tumor site [51]. To overcome the clearance effects of AETs and promote the transport of anticancer agents to across the BBB, the receptor-mediated transport process has to be active through binding of the drugs to a cell-surface receptor.

GBMs compromise the integrity of the BBB and result in a highly heterogeneous vasculature with distinct features of nonuniform permeability and active efflux of molecules. This phenomenon is known as the blood-tumor barrier (BTB) [52]. Both the BBB and BTB limit the access of potentially effective chemotherapeutic agents to metastatic lesions. Recently, numerous strategies have been investigated to overcome these barriers, including new small molecules capable of penetrating the BBB, novel formulations of anticancer agents, and various disruptive techniques [52,53,54]. A drug-loaded nanocarrier has been designed to overcome the BBB and BTB through the increased affinity for an endocytic receptor expressed on the endothelial cell surface, leading to the efficient release to tumor sites [55,56].

### 2.3. Applicable Strategies for Drug Delivery to GBMs

There are various strategies to enhance most drugs to cross the BBB, including chemical modification of anticancer drugs, strategies to increase the BBB permeability, and efflux transporter inhibitors [55,57]. For example, increased solubility and lipophilicity of methotrexate can be achieved by adding the translocator protein to form the TSPO-MTX conjugates, which will enhance their delivery through the BBB [58]. A targeted ultrasonic wave can be used to transiently alter the permeability of the BBB through an interaction between administered microbubbles and the capillary bed [59]. MRI-guided focused ultrasound has been demonstrated to open the BBB in the targeted region without compromising its histological and functional integrity [59,60]. Recently, a low dose of systemically injected recombinant human vascular endothelial growth factor was shown to help induce a short period of increased BBB permeability [61,62].

Furthermore, the efflux transporter inhibitors have been used in a mouse model to investigate the effects of improving drug delivery across the BBB [63]. Both P-glycoprotein (Pgp, also known as ABCB1 or MDR) and breast cancer resistance protein (BCRP, also known as ABCG2) are well known efflux transporter proteins found on the endothelial cells of the brain that present an additional functional barrier by pumping drugs back into the blood circulation [64]. Pgp is also highly expressed on the BBB and GSCs and limits the therapeutic efficacy of several chemotherapeutic drugs targeting GSCs. Therefore, safe inhibitors of Pgp, including thiosemicarbazone derivatives and tetrahydroisoquinoline derivatives, can bypass Pgp-mediated drug efflux in primary human BBB and GSC cells [65]. The limitations of efflux transporter inhibitors include poor bioavailability and varied permeability depending on the drug or molecules measured and the heterogeneity of GSCs [66]. For example, statins can reduce the efflux activity of Pgp and BCRP by increasing NO synthesis, which have been documented in statins plus doxorubicin-loaded nanoparticles to be efficient vehicle to cross the BBB [67].

Another strategy to overcome BBB-associated drug delivery is continuous local drug delivery using a convection-enhanced delivery (CED) to achieve great distribution within the brain [68]. In cases of diffuse tumor infiltration and inability of curative surgical resection, CED is also applicable by facilitating concentrated therapeutic drug delivery regardless of molecular size and charge [68,69]. The implantable reservoir-catheter system is the basic CED that uses a pump to provide continuous positive pressure for local drug delivery. In addition, a CED has the advantages of delivering a diverse range of chemotherapeutic agents, monitoring the volume of distribution, and inducing almost no systemic toxicity [69]. However, optimal drugs for CED require specific considerations, including limited toxicity to the normal brain and tumor cell specific cytotoxicity, and a long therapeutic half-life [70]. In addition, CED causes white matter edema, active tumor/BBB disruption, backflow through the catheter, and air bubbles, all of which need technical improvements to facilitate the application of CED to treat GBMs [70].

## 3. Nanocarriers for Delivery of Anticancer Agents

### 3.1. Basic Concept and Characteristics of Nanocarriers

Because only a small number of molecules can cross the BBB, novel technologies and delivery systems are therefore necessary to efficiently transport the drugs into the brain matrix. Nanocarriers and nanotechnology-based delivery of drugs are colloidal-based particulate systems that can overcome the BBB due to their characteristics of biosafety, sustained drug release, increased solubility, enhanced drug bioactivity, BBB penetrability, and self-assembly [55,56].

Chemotherapeutic agents are entrapped inside the matrix or attached to the surface of nanoparticles, which are capable of penetrating small capillaries because of their small size. After extravasation and receptor-mediated transcytosis, nanoparticle-drug complexes are absorbed by cells; then, the drug is released into their cytoplasm or compartment (Figure 3). After penetrating the BBB, the accumulation of chemotherapeutic agents-loaded nanoparticles in tumor sites is influenced by the interaction of nanoparticle with tumor cells and intra-tumoral diffusion, which is significantly affected by the particle size, morphology, and surface properties of the nanoparticle [71,72]. Nanoparticles from biodegradable materials have the most important advantage of sustained drug release at the targeted site in a tunable manner [73,74]. Through appropriately engineering with proper ligands on the surface, the drug-loaded nanoparticles can be nontoxic, nonimmunogenic, and stable inside the blood circulation [75]. The presence of ligands on the surface of nanoparticles can deliver the carrier system to the target sites with specific receptors [76,77]. Good candidates to be the ligands that enable drug-nanoparticle complex to efficiently pass through the BBB via receptor-mediated endocytosis include transferrin, apolipoprotein (Apo) E, B, A and some antibodies on the surface of nanoparticles [76,77,78,79]. In addition, nanotechnology can improve the bioavailability of short half-life chemotherapeutic agents and reduce the adverse side effects through the combination [80].

The size and surface charge of nanoparticles contribute significantly to their ability of escape from the reticuloendothelial system (RES) [81]. Nanoparticles with a size between 5 nm and 500 nm and a positive charge is very important for better cellular uptake. Particles < 200 nm are especially preferred and suitable for systemic administration [81,82]. Particles cannot be less than 5 nm because they are easily excreted by the kidney. The reason for preferred positive charge nanoparticles is that the better interaction with negatively charged cell membranes, target biological area, or some proteins enhance the in vivo stability in the circulation [82]. The small size of nanoparticles is also the double-sided blade because nanoparticles can enter the cytoplasm of normal cells after crossing the tissue junctions and cellular membranes. Inside the cells, nanoparticles can induce mitochondria structural damage, cause damage to DNA and RNA, and lead to cell death [83]. To make the nanoparticles applicable in clinical medicine, surface coatings and other modifications to increase the safety of nanoparticles in the body are mandatory.

To minimize unwanted interactions between nanomaterials and normal tissues, surface modification of nanoparticles with different molecules has been investigated for more than a decade [83]. Initially, polyethylene glycol (PEG) was used as a surface coating because of its hydrophilic external surface and inner hydrophobic polymeric matrix, which helps nanoparticles escape RES recognition and increases the half-life and persistence in the circulation [84]. In order to increase the affinity and specificity of nanoparticles for the targeted tissue, chitosan PEGylated albumin coated nanoparticles coupled with some antibodies were later developed for brain drug targeting through receptor-mediated transporter endocytosis [85,86]. Although there have been various drug delivery systems to the CNS [59,60,61,62,63,64,65,66,67,68,69,70], a recently developed nanoparticles from poly (ethylene glycol)-poly(ῳ-pentadecalactone-co- p-dioxanone) can have a longer period of sustained release and no requirement of repeated infusions, which enhances safety and translatability [87].

### 3.2. Applicable Strategies of Nanocarriers to Improve Delivery of Anti-GBM Drugs

The major obstacles of GBM treatment are the presence of the BBB, the capture and clearance of anticancer agents by the RES, and the lack of a specific targeting mechanism by which the drugs can bind specifically to GSCs. Special designs and administration route of nanocarrier-based delivery systems are desperately needed to overcome these obstacles. Through CED and an intratumor administration route, nano-formulated drugs can be maintained in or around the tumor site for a longer period, which cannot be achieved in non-nano-formulated drugs [88]. The technique of CED has additional advantages of allowing nano-formulated chemotherapeutic agents to be released toward GBM cells at a precisely controlled infusion rate [68,89], which further enhances anti-tumor efficacy [89].

A new synthesized nanoparticle from magnetotactic bacteria was recently intratumor injected in mice bearing intracranial glioma and followed by alternating magnetic field or magnetic hyperthermia, which showed enhanced anti-tumor efficacy with almost full tumor disappearance [90,91]. This approach indicates an available and alternative strategy for the treatment of infiltrating tumors such as glioma. The whole tumor coverage by nanoparticles is difficult to be achieved.

Another method is to inject anticancer drugs via the intranasal route. For example, the potential of the nose-to-brain direct transport, which bypasses the BBB, has been investigated in GBM mice using theranostic polyfunctional gold-iron oxide nanoparticles surface loaded with therapeutic miRNAs [92]. This nanoformulation also allows GBM cells to be systemically delivered to TMZ [92]. The intranasal route of nose-to-brain drug delivery can potentially present several advantages over the traditional IV route. However, this delivery system is mostly in a preclinical phase of development, and intranasal administration also has limitations [93]. Lower bioavailability of peptides and high clearance from the nasal cavity and some restrictions from the anatomy of the nasal cavity are currently obstacles that need to be overcome [93].

It is possible to weaken or open the major barrier by MRI-guided focused ultrasounds or administering bradykinin, which enable the chemotherapeutic agents to diffuse through the BBB more efficiently [94,95]. Cisplatin-loaded nanoparticles coated with PEG, which prevent capture by macrophage, can have brain-penetrating ability to cross the BBB and BTB after MR image-guided focused ultrasound [60,94]. The successful combination in animal models may offer a new powerful approach for treatment of refractory GBM and control of recurrence [60].

AET targeted and tight junction targeted strategies are important methods to achieve the goal of circumventing and modulating the BBB and BTB [96]. Previously, inhibitors against multidrug resistance efflux transporters have failed in most studies. However, Pgp inhibitors encapsulated into surfactant-based nanoparticles have been developed to reverse multidrug resistance efflux transporters, which can be used to improve the therapeutic effect of the drug [97].

A magnetic field has been applied to trigger the diffusion of magnetic anti-GBM drugs towards the GBM cells [90,91]. Previous difficulties came from the unavailability of equipment to generate a sufficient and precise magnetic field and concerns of unwanted influences on normal tissue [98,99]. However, direct intratumoral administration of magnetic nanoparticles (MNPs) is now applicable for GBM treatment because MNPs can be highly accumulated to the tumor site after the development of a magnetic platform drivable through an external magnetic field [100]. Another example is the new design of hybrid magnetic nanovectors, which are angiopep-2-functionalized lipid-based and promote GBM cell death through a combined effect of lysosomal membrane permeabilization and chemotherapy [101].

### 3.3. Targeting the GBM Cells and Glioblastoma Stem Cells

The current strategy of active targeting for GBM uses substances attached to the surface of nanoparticles that can specifically target the receptors or antigen on GBM cells or GSCs [89,102,103]. GBM cells express several receptors or proteins, such as metalloproteinase-2, IL-13 receptor, Integrinα5β3, CD33, and CD133, which can be the candidate for nanoparticle targeting.

Since the presence of GSCs account for an important cause of GBM recurrence, the important targeting of GSCs has been investigated in recent years [104]. GSCs express several specific receptors or markers that can be the target of nanocarrier-based drug delivery system. Based on the locations, GSCs have cell surface markers (e.g., CD15, CD133), transcription factors (e.g., OCT4), post-transcriptional factors, and cytoskeletal proteins (e.g., nestin) [105]. The majority of treatments to target GSCs have failed in clinical trials, despite a number of treatment options for targeting GSCs being theoretically available [106].

Recent examples of GSC targeting using nanotechnology includes the preparation of mixing calf thymus DNA with gold nanoparticles, which sensitizes GSCs to radiotherapy [107]. The neurofilament-derived NFL-TBS.40-63 peptide and LinTT1 peptide with enhanced binding targets GSCs [108,109]. Nestin positive GSCs can also be specifically recognized by gold nanorods functionalized with an engineered peptide, which has been proven as a promising tool to develop an efficient nanomedicine for treatment of recurrent GBM [110]. For cell surface marker CD133 in GSCs, the targeting peptide CBP4-coated gold nanoparticles has been developed as a drug carrier for therapeutic approaches [111].

## 4. Current Nanocarriers and Nanocarrier-Associated Strategies for the Treatment of GBM

The new development of nanocarrier-based combination therapy for GBMs has additional advantages, including facilitation of sequential drug exposure, well confirmation of the synergistic drug ratio, and improved localization of anticancer agents into the tumor site [55,57]. Nanocarriers can be classified into nanocapsules, nanoparticles, and nanospheres depending on their preparation methods. Among them, nanoparticles are the most widely used to treat GBMs and can be classified based on the type of colloidal drug carriers from which they are made of, including liposomes, polymeric nanoparticles, solid lipid nanoparticles, polymeric micelles, silica, and dendrimers.

### 4.1. Liposomes

The structure of liposomes is similar to that of cell membrane, as they are composed of a water soluble core surrounded by an outer phospholipid membrane. This characteristic increases lipophilicity and enables lipophilic macromolecules to cross the BBB. Liposomal nanoparticles have a lot of advantages, including easy preparation, easy encapsulation of a wide range of anticancer drugs, favorable biocompatibility, efficiency, non-immunogenicity, improved solubility of anticancer agents, and commercial availability [112,113]. Liposomes were initially designed to encapsulate radiosensitizers and chemotherapeutic agents such as doxorubicin for the treatment of various refractory cancers for more than two decades ago [112]. During the last decade, various methods of liposomal formulations for the treatment of GBMs, novel conjugated agents, and receptor-mediated transcytosis have been investigated to facilitate their transport across the BBB [113,114,115]. For example, conjugation of polyethylene glycol (PEG) to the surface of a liposome phospholipid bilayer can extend the half-life of liposomes in the circulation because PEG can help the nanoparticles escape from the capture of RES [84].

Some unique receptors or antigens overexpressed on GBM cells are the potential tumor targets for the development of novel nanotechnology. For example, interleukin (IL)-13-conjugated liposomes and IL-4 receptor-targeted liposomal doxorubicin have been investigated in mouse models, which showed evidence of significant tumor size reduction when compared with unconjugated liposomes [114,115]. This approach does not increase toxicity in animals receiving receptor-conjugated liposomes [114], indicating it as a potential application of nanotechnology. Furthermore, an antibody can be used to label liposomes to target tumors. Anti-EGFR immunoliposomes were developed more than ten years ago to target GBM cells with overexpression of EGFR in an animal model and demonstrated that they can significantly enhance the efficacy of multiple anticancer drugs [116].

Despite the common application of liposomal nanoparticles in GBM treatment, there are some disadvantages we need to overcome. Non-uniform effects across all brain regains are noted in liposomal nanoparticles, and its permeability across the BBB varies depending on the loaded drug or surface molecules [114,115].

### 4.2. Polymeric Micelles

Polymeric micelles are composed of a hydrophobic polymer core and hydrophilic shell architecture. This architecture is formed through the self-assembly of block copolymers, and the design can modulate the incorporation efficiency and controlled release rate of chemotherapeutic agents [117]. The characteristic core-shell structures and narrow size distribution of 10–100 nm can effectively protect the drug-loaded core from interaction with the complement system and macrophage uptake, which contributes to their prolonged circulation with a long half-life of more than 10 h [118,119]. Poly (caprolactone), poly (D,L-lactide), poly (D,L-lactide-co-glycolide), and long-chain alkyl derivatives are biodegradable polyesters and commonly used as the core-forming polymer [117]. Poly (ethylene glycol) (PEG) is the ideal shell-forming polymer, which can avoid interaction with serum proteins [117,118].

After resolving previous weakness of inadequate drug-circulation time, the major obstacle to the implementation of polymeric micelles-based GBM therapy is the lack of targeting moieties that could allow for greater GBM specific accumulation [117]. Therefore, further effects of targeting specific receptors expressed on GBM cells are ongoing to improve the efficacy of current formulation. For example, polymeric mixed micelles composed of Pluronic P-123 and F-127 containing 17-Allylamino-17-demethoxy geldanamycin (17-AAG) can be a good nanomaterials-based drug delivery carrier because 17-AAG is a potent inhibitor of heat shock protein 90 (Hsp90) and can cause destabilization of Hsp90 related client proteins in cancer cells [119,120]. The design of 17-AAG loaded Pluronic P-123 and F-127 mixed micelles is favorable, and the targeting ability of 17-AAG, controlled release rate and high drug loading have also been documented as a potential delivery system for GBM treatment [119]. Transferrin receptor (TfR) is the promising target site because it is overexpressed on both the BBB and GBM cells. Sun et al. designed the TfR-PEG polymeric micelles, which could be absorbed rapidly by tumor cells, and traversed effectively the BBB [121]. TfR-PEG polymeric micelles loaded with paclitaxel can effectively inhibit the proliferation of U87 GBM cells in vitro, and prolong median survival of nude mice bearing GBMs [121].

### 4.3. Dendrimers

Dendrimers are the smallest molecules with sizes less than 12 nm and have highly branched and compact scaffolds architecture, which is suitable for transporting short interfering RNA (siRNA) and protecting it from degradation in the circulation [122,123]. Additional advantages of dendrimers loaded with methotrexate include increased drug potency and high efficiency of crossing the BBB [124]. However, dendrimers also have some disadvantages, such as rapid clearance of the RES, toxicity to normal tissue because of interaction with cell membrane, and relatively poorer controlled release behavior [122,125]. Therefore, numerous functionalized strategies, such as attachment of lipid, amino acid, peptide, or aptamer, have been used for modification of dendrimers [123,126].

Recently, poly (amidoamine) (PAMAM) dendrimer-entrapped gold (Au) nanoparticles has been prepared to compact two different siRNA for oncogene silencing. In the newly novel approach, the PAMAM-Au dendrimers are coated with beta-cyclodextrin (β-CD), which has been demonstrated to be efficient carrier for delivery of siRNA to glioma cells [125,126]. The endogenous amino acids improve the biocompatibility and endosomal escape of amino acid functionalized dendrimers, while phosphate dendrimers with hydrophobic backbone and hydrophilic surface can have better penetration through the BBB [127,128]. Another example is the arginine-glycine-aspartic functionalized dendrimer-entrapped gold nanoparticles, which have good cytocompatibility and highly efficient transfection capacity and have been demonstrated as potentially efficient gene therapy for GBMs [127]. Polyether-copolyester (PEPE) dendrimers conjugated with d-glucosamine have been designed to enhance the drug delivery across the BBB and tumor targeting [124]. The in vitro model has showed that glycosylation of the PEPE dendrimers not only increase the rapid accumulation around the tumor spheroids but also overcome MTX resistance because methotrexate-loaded glucosylated PEPE dendrimers was able to kill even MTX-resistant cells [124].

### 4.4. Metal Particles

Metal particles can enhance radiosensitization of GBM tumor cells and significant DNA damage of tumor cells have been observed in animal models treated with metal particles prior to radiation therapy [129]. The metal particles own characteristics of high X-ray absorption, synthetic versatility, and unique electronic properties, which accounts for their good candidates as radiosensitizers [130]. Among noble metal inorganic nanoparticles, gold nanoparticles (AuNPs) are characterized by easy modification, controllable diameters, and large surface/volume ratios, and are one of the most ideal nanomedicine materials for GBM therapy [131]. The controlled size of AuNPs makes it easily cross the BBB, but its clinical application is limited by lack of targeting ability [131,132].

Recently, a DNA aptamer selected from a large random single-stranded DNA has been prepared to target EGFRvIII of GBMs [132]. The targeting efficiency of aptamer is further enhanced by entrapped into AuNPs through a gold-sulfur covalent bond [132]. The aptamer-AuNP complexes have been demonstrated as a new type of drug candidate for GBM therapy because they showed efficient antitumor effects in vivo and in vitro inhibition of tumor proliferation [132]. The weak transmembrane penetration of aptamer is overcome by the appropriately sized AuNPs. Nanoparticle can also help delivery of the therapeutic gene targets. A novel polyfunctional gold-iron oxide nanoparticle to deliver therapeutic miR-100, the tumor suppressor, was recently designed and proved to enhance sensitization of GMB cells to the systemically administered TMZ in mice [92].

Previous concerns of metal particles include their cytotoxicity and physical damage to the normal tissue after long-term accumulation in the circulation [133]. The mechanisms of metal particle toxicity include induction of oxidative stress, inflammatory cytokine release, lysosome degradation, and DNA destruction [133,134]. However, several gold and silver nanoformulations entrapped with chemotherapeutic agents are already approved by the American Food and Drug Administration for clinical trials, since their biodistribution and mode of clearance are now well understood [135].

### 4.5. Silica

Silica nanoparticles (SiNPs) have several benefits commonly used in various medical applications, including good biocompatibility, large surface area for drug loading, stability, and inexpensive costs [136]. Previous concerns of their cytotoxicity, DNA destruction, and production of reactive oxygen species limit the clinical application of SiNPs as biomarkers, cancer therapeutics, or drug delivery system [136,137]. Later SiNPs have been investigated in many research areas for their clinical safety and potential applications. Because the SiNPs-induced toxicity can be controlled by appropriate size, dose, and cell type [137,138], researchers now can try multimodal modifications of SiNPs to make it clinically applicable. The greater toxicity of smaller-sized SiNPs can be modified by synthetic modification of SiNPs [139].

For GBM treatment, transferrin-modified porous silica nanoparticles are current popular formulation, which can have high biocompatibility, degradability, and high drug-loaded capacity [140,141]. The transferrin-functionalized pSiNPs can achieve a sustained release of the drug (such as doxorubicin) at the targeted site because transferrin receptor is often overexpressed on the BBB and the surface of GBM cell only. A multicomponent nanoparticle composed of a mesoporous silica shell and an iron oxide core with fibronectin-targeting ligands has also been developed, which can have an efficient, large amount, and widespread drug delivery into the GBM after an external low-power radiofrequency field [142].

### 4.6. Nanoparticle-Induced Hyperthermia

The combination of hyperthermia and modern radiation and/or chemotherapy has been used for nearly half a century. The mechanisms of hyperthermia-induced radiosensitization and chemosensitization include impaired DNA repair, increased apoptotic pathways, heat-induced inhibition of the AKT signaling pathway, and disruption of the BBB [143,144,145]. Local temperatures up to 45 °C induce GMB cell apoptosis in a murine animal model [144]. Although various techniques, including radiofrequency, ultrasonic waves, water baths or heat blankets, microwaves, laser-induced interstitial thermotherapy, and magnetic nanoparticles (MNPs)—have been used to exert hyperthermic effects on tumors, MNPs have the advantages of direct intratumoral administration, high localized accumulation to create sufficient heat generation in tumors, and good efficacy [144].

MNPs are ideal candidates for CED application for the treatment of GBMs. Real time MRI-guided MNP delivery into the brain via CED has been investigated for decades [69,146]. Iron oxide MNPs are preferred for magnetic hyperthermia due to their high heating capacity and have been designed to therapeutically target cancer cells [146]. Recently, the targeting effects of iron-oxide nanoparticles conjugated with the EGFR inhibitor, cetuximab, were found to have a significant antitumor effect in EGFRvIII-expressing GSCs [146]. Fan et al. also demonstrated a novel theranostic complex of superparamagnetic iron oxide-loaded microbubbles for drug delivery to the brain, and the distribution and quantitative deposition of the agent was also accurately estimated [147].

Although the safety and effectiveness of MNPs can be confirmed by an accurate and reliable treatment plan, the heterogeneous response to magnetic hyperthermia within the GBM mass limits their clinical applications. For example, a transient increase in the growth of the CD133 subtype of gliomas after hyperthermic preconditioning was noted in a recent xenograft model [148]. Furthermore, the issue of MNP toxicity deserves to be further investigated and depends on the chemical composition, surface coatings, physical characteristics of MNPs, and local concentration. For example, MNPs containing iron oxide and titanium are less toxic than those composed of heavy metals, including gold, silver, cobalt, zinc, and cadmium. Recent researchers made use of dextran and bovine serum albumin as the surface coatings of MNPs, which have been demonstrated to reduce toxicity and prevent intravascular coagulation [144,149].

### 4.7. Nanoparticles as Carriers of Antitumor Antibiotics

Various chemotherapeutic agents including doxorubicin, bleomycin, epirubicin, daunorubicin, and actinomycin D are classified as antitumor antibiotics because they are produced by *Streptomyces* bacteria and cause cell death by interfering with DNA replication and damaging DNA in GBM cells [150]. These antitumor antibiotics show great antitumor effects on GBM cells in vitro, but are ineffective in vivo due to their poor ability to penetrate the BBB [50]. The resolution is to encapsulate these chemotherapeutic agents in PEGylated liposomes and apply effective drug delivery strategies [61]. For example, the loading of doxorubicin in poly (lactide-co-glycolide) nanoparticles coated with poloxamer 188 (Dox-PLGA) enables enhanced brain delivery [61,151]. Another example is ultrasound-induced microbubbles, which effectively deliver drugs, such as liposomal doxorubicin, to enter the brain through a transient opening of the BBB in a rat glioma model [152].

## 5. Clinical Trials

Although numerous in vivo and in vitro studies have been conducted to prove the efficiency and therapeutic potential of nanotechnology and/or nanocarriers-based treatment of GBM, few clinical trials using nanotherapies to target GBM have been completed. The information about clinical trials focusing on GBM treatment is summarized in Table 1.

A combination of TAZ and pegylated liposomal doxorubicin (PEG-Dox) has been commonly used as the post-operative treatment for newly diagnosed GBM patients following chemo-radiotherapy and also for patients with high grade recurrent GBMs since nearly two decades ago [157,158,159,160,161]. These clinical trials found that liposomal doxorubicin was tolerable and feasible, with the main side effects being palmaroplantar erythrodysesthesia and myelosupression. The 12-month progression free survival after combined TAZ and PEG-Dox regimen was reported to range between 15–30.2% and the median overall survival was 13.4–17.6 months. Although combination of TAZ and PEG-Dox is well tolerated, it does not add significant benefit regarding patients’ outcomes [157,158].

The EnGenelC delivery vehicle (EDV) is a novel nanocellular (minicell) compound that can encapsulate adequate concentrations of chemotherapeutic agents to target EGFR, overexpressed in 40–50% of patients with GBMs [154]. Therefore, a phase I study of EDVDox (EDV containing doxorubicin) was conducted in 14 GBM patients. This new regimen is well tolerated with only nausea, fever, and chills or rigors experienced in some of patients [154]. No previous side effects of palmaroplantar erythrodysesthesia and myelosupression were observed in these patients. Although all of these combination regimens are effective and well tolerated in clinical trials, none has been documented to significantly result in a meaningful improvement of a patient’s outcome [154,157,158,159,160,161]. It can be hypothesized that these nanoformulations would help to improve GBM treatment, but the clinical significance is limited by inadequate case numbers.

Hyperthermia can increase the cytotoxic effects of radiotherapy. Therefore, magnetic iron-oxide nanoparticles are used and directly injected into the tumor and subsequently stimulated by an alternating magnetic field to generate heat [153]. This approach has been demonstrated feasible and effective in animal models [144,146], and later in clinical trials [153,162]. In Europe, a MNP compound (NanoThem^®^ AS1; MagForce Nanotechnologies AG, Berlin, Germany) for magnetic hyperthermia application in combination with radiotherapy for patients with recurrent GBM has been approved [154]. However, there are several obstacles for clinical application of MNPs, including the difficulty of accurate intratumoural heating and precise temperature control at the GBM site, the presences of pacemakers and defibrillators as the contraindication, and the removal of dental filling, implants, and crowns [153,162].

Currently some new nanoformulations, including EnGenelC delivery vehicle (EDV)-doxorubicin, 5-fluorouracil-releasing microspheres, or Semliki Forest virus vector carrying IL-12 gene encapsulated in cationic liposomes have been proven safe and efficient in GBM patients [154,155,156], but the beneficial effects of these regimens require further clinical trials. In addition, a phase II study of combined TAZ and targeted P53 gene therapy (SGT-53, previously conducted on other solid tumors [163]) for treatment of patients with recurrent GBM is currently recruiting participants.

## 6. Conclusions

GBMs are well known for the poor prognosis and current therapeutic strategies have not improved overall survival or progression free survival. Because of their characteristic size, shape, and surface properties, nanoparticles are capable of encapsulating and delivering therapeutic molecules to the brain. In clinical trials, these nanoformulations combined with oral TAZ and radiotherapy have been used in GBM patients after maximal resection and are well tolerated in most patients. The MNPs for hyperthermia allow for a tumor-specific and sustained effect, and GBM cells can be sensitized to radio-chemotherapy through hyperthermia effects. However, none of these regimens do significantly improve patients’ outcome in terms of progression free survival or overall survival. The limitation of these clinical trials may be due to a lack of large number or at least sufficient patients to reach statistical significance.

Future direction of nanotechnology and clinical applications may consider monoclonal antibodies, combining GSC-targeting SGT-53 with traditional TMZ, or novel nanoformulations loaded with therapeutic miRNAs to improve immunotherapy and antiangiogenic processes. MNPs are the promising nanoparticles for intratumoral hyperthermia therapy in patients with GBM. However, most effects of various nanoformulations on GBM cell models cannot be replicated in actual clinical trials because tumor heterogeneity remains unpredictable and the major obstacle for successful GBM treatment.

## Figures and Tables

**Figure 1 cancers-13-00195-f001:**
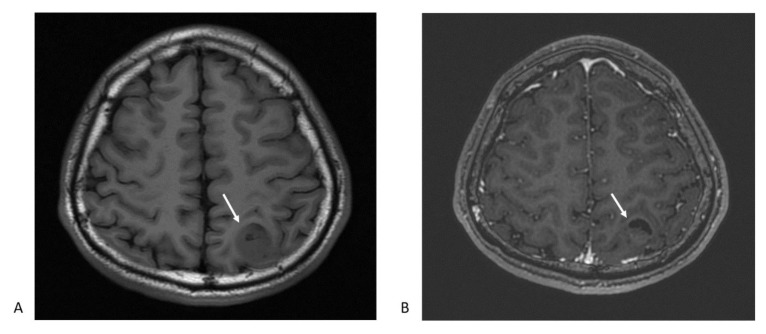
Magnetic resonance imaging (MRI) typical features glioblastoma multiforme: (**A**) The axial T1-weighted image shows heterogeneous hypointense mass lesion at left parietal lobe (arrow); (**B**) postcontrast T1-weighted axial image depicts an enhancing ring lesion with central heterogeneous enhancement. The crescent-shape dark area suggests a necrotic part of the tumor (arrow).

**Figure 2 cancers-13-00195-f002:**
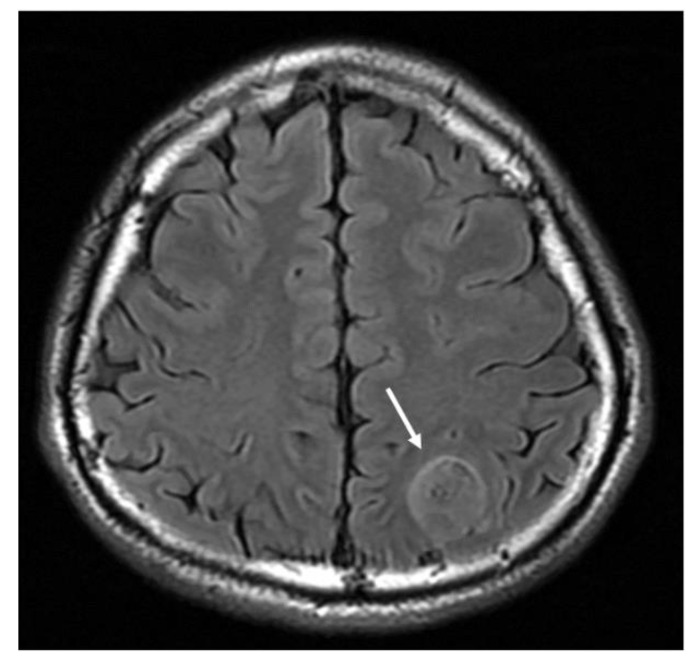
MRI T2-weighted fluid attenuation inversion recovery (FLAIR) axial image demonstrates a hyperintense mass lesion (arrow).

**Figure 3 cancers-13-00195-f003:**
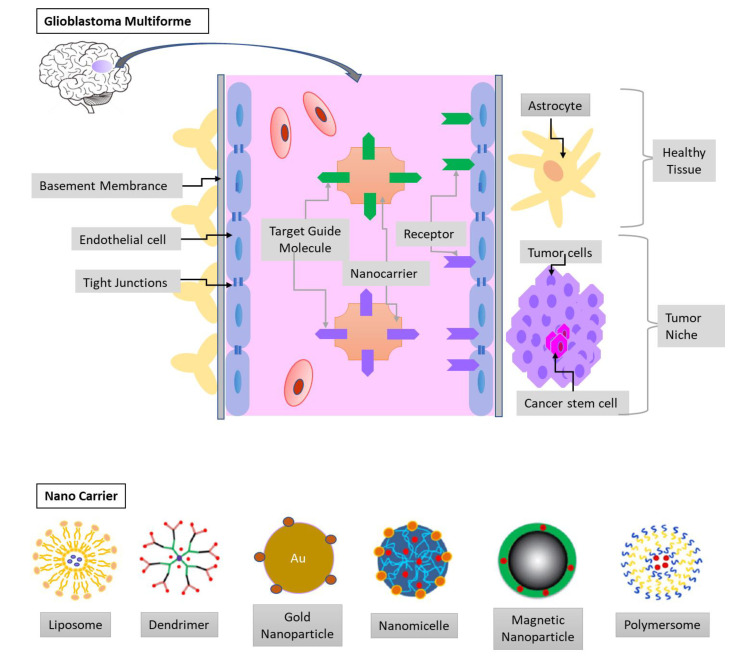
The blood-brain barrier (BBB) and the glioblastoma multiforme (GBM) niche. The endothelia cells, tight junctions, and basement membrane limit the drug delivery to the tumor niche. Therefore, nanocarrier conjugated with target guide molecules and loaded with chemotherapeutic agents can efficiently cross the BBB. The composites of nanocarrier can be liposomes, micelles, dendrimers, metal, and polymeric nanoparticles.

**Table 1 cancers-13-00195-t001:** Clinical trials using nanotechnology and nanocarrier-based delivery systems for treating glioblastoma multiforme.

Trade Name/References	Case Number/Patients	Formulation/Composition	Main Results
Nanothermotherapy Phase II/[153]	59 patients with recurrent GBM	Thermotherapy and Magnetic iron-oxide nanoparticles + reduced dose radiotherapy	This combination is safe and effective, leading to longer overall survival.
EDV-doxorubicin Phase I/[154]	14 patients with recurrent GBM expressing EGFR	EnGenelC delivery vehicle (EDV)-doxorubicin + radiation and oral TMZ	EDVDox was well tolerated, with no dose limiting toxicity and no withdrawals from the study due to adverse events.
Interleukin-12 Phase I, II [155]	Adult patients with recurrent GBM	Semliki Forest virus vector carrying IL-12 gene encapsulated in cationic liposomes	Liposomally encapsulated virus can be efficiently delivered to GBM using the convection-enhanced delivery.
5-fluorouracil Phase II/[156]	95 GBM patients were randomized after surgery	5-fluorouracil-releasing microspheres followed by early radiotherapy	Only slightly increased overall survival in the study group when compared with those received radiotherapy alone.
Caelyx, PEG-Dox Phase I, II/[157]	63 patients with newly diagnosed GBM	Pegylated liposomal doxorubicin + prolonged TMZ and radiotherapy	The progression free survival after 12 months was 30.2%, and the median overall survival was 17.6 months. Neither the addition of PEG-Dox nor prolonged temozolomide resulted in a meaningful improvement.
PEG-Dox Phase II/[158]	40 patients with newly diagnosed GBM after surgery	TMZ and Pegylated liposomal doxorubicin after radiotherapy and surgery	The progression free survival after 6 months was 58%, and the median overall survival was 13.6 months. Combination of temozolomide and PEG-Dox does not add clinical benefit.

TMZ: temozolomide; PEG-Dox: Pegylated liposomal doxorubicin; EDVDox: EnGeneral delivery vehicle-doxorubicin; GBM: glioblastoma multiforme; EGFR: epithelial growth factor receptor; IL-12: Interleukin-12.

## Data Availability

The datasets used/or analyzed during the current study available from the corresponding author on reasonable request.

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
