# Peer review of "Nanotechnology and Nanocarrier-Based Drug Delivery as the Potential Therapeutic Strategy for Glioblastoma Multiforme: An Update"

_cancers, 2021, doi:10.3390/cancers13020195_

Round 1
Reviewer 1 Report
please see the attachment

Author Response
RE: Cancers-1037486
Nanotechnology and nanocarrier-based drug delivery as the potential therapeutic strategies for glioblastoma multiforme: an update
Dear Editor,
Thank you for your appreciated comments on our manuscript. We had the manuscript revised, all according to the reviewers’ and editor’s suggestions. We underline every change and highlight in red color on the revised manuscript. The replies for the reviewers’ criticisms are as followings. We hope this revised version can be acceptable.
Best regards,
Ming-Horng Tsai
Chief, Division of Neonatology and Pediatric Hematology/Oncology, Department of Pediatrics, Yunlin Chang Gung Memorial Hospital, Taiwan, R.O.C.
Comments from Reviewer No.1 :
The article by Hsu et al. entitled ”Nanotechnology and nanocarrier-based drug delivery as the potential therapeutic strategy for glioblastoma multiforme: an update’’ reviews the latest advances in anti-glioblastoma therapy based on nanotechnological approach. Although I like the overall concept of the manuscript, I think it is not entirely well executed in terms of detailed description of modern “nano-based’’ oncopharmacology. On the plus side, I like that current literature is featured here (approx. 85 % is less than 10 years old), and the subsection 5. Clinical Trials, which is actually my favorite one of the whole manuscript, is very well written. However, although I think that this manuscript raises an interesting subject and Authors made an effort to write it, I also have some major concerns about this manuscript.
My detailed comments are given below:
Major points:
- The main issue with this manuscript in my opinion is that the introduction part is very redundant and not entirely connected with the overall topic of the manuscript. In my opinion 1. Introduction and 2. Obstacles of GBM treatment and the resolution sections should be condensed into one concise Introduction section to delineate only the most important facts about GBM. Although I appreciate the detailed description of GBM pathophysiology, and I find Fig.1 and Fig.2 very nice from the medical point of view, it seems irrelevant from the perspective of the topic discussed herein. If structural characteristics of GBM were further used to discuss the advances of “nano-based’’ imaging or therapeutic approaches, then yes, but without “nano’’ context it seems out of place here.
Reply:
Thank you for your appreciated comments and instructive advice. I will condense the 1. Introduction and 2. Obstacles of GBM treatment and the resolution sections in the revised manuscript. Because I have searched several relevant articles and most of which have these two sections apart, I will still arrange these into two sections in the revised manuscript but I condense them, thank you.
For figure 1 and figure 2, I appreciate the reviewer like them, and I beg to keep them although I agree with the reviewer’s opinion that it seems irrelevant from the perspective of the topic discussed herein. However, two of the co-authors had contributions to this manuscript because they provided the MRI imaging, figures and information of this manuscript. If I deleted them, then it is not reasonable to list them as the co-authors. Therefore, I beg to keep this part.
Please see the revised manuscript, I have provided a lot of more information focusing on the topic, then the first two sections can be just the basic information of GBM.
- Also, as far as I consider the “nanotechnological’’ aspect of this manuscript should be added to this work. In the present form, there is only brief “teaser’’ describing development of nanotechnology in the field of brain malignancies in section 3. Nanocarriers for delivery of anticancer agents. More context should be put in terms of nanomedicine of GBM.
Reply:
Thank you for your instructive advice. I will add more nanotechnological development for nanomedicine of GBM in the section 3. Nanocarriers for delivery of anticancer agents. (all the red color words with lines from line 226- line 330, and relevant references no.72-115).
- No silica nanoparticles are mentioned here and considerable number of reports have recently been published concerning utilization of these particles in GBM e.g. Turan O., et al. Adv Ther (Weinh) (2019), Kusaczuk M., et al. Int J Nanomedicine (2018), Sheykhzadeh S., et al. Sci Rep (2020) etc.
Reply:
Thank you for your instructive advice. I will add silica nanoparticles and relevant references in the revised manuscript. (line 442-459, and references no. 140-146).
- I also think that the nanoparticle-connected subsections 3.1-4.2 are a little bit underdeveloped. I miss the clear concept of these subsections. If it was supposed to be presented from strictly technological point of view, it is not comprehensive enough. On the other hand, if it was supposed to be of more medical context, it lacs the basic facts about molecular and cellular effects of particular “nano-treatment’’ in GBM.
Reply:
Thank you for your instructive advice. It is supposed to be of more medical context, and I will add basic facts about molecular and cellular effects of particular “nano-treatment’’ in GBM in subsection 3.1 and 3.2. (line 226-266)
- I think that the 4.1 and 4.2 subsections should be renamed to gain any nanoparticle context. Please consider Nanoparticle-induced hyperthermia for 4.1., and Nanoparticles as carriers of antitumor antibiotics for 4.2 (or something similar).
Reply:
Thank you for your instructive advice. I will revise it as “Nanoparticle-induced hyperthermia” for 4.6., and “Nanoparticles as carriers of antitumor antibiotics” for 4.7 accordingly, thank you.
Minor points:
- Although the manuscript is written in overall good English, authors did not avoid certain grammatical and stylistic mistakes, e.g.:
- line 23: I do not recommend using plural of “glioblastoma multiformes’’ please stick with glioblastoma multiforme (GBM)
Reply:
Thank you for your instructive advice. I have revised as “glioblastoma multiforme (GBM)” accordingly. (line 23)
- line 37: ”patient’s outcomes’’, Saxon genitive is wrongly placed
Reply:
Thank you for your instructive advice. I have revised as “patients’ outcomes”
- line 54: ,,(…) although their different genetic bases and molecular pathways underlying tumorigenesis are quite different.’’
Reply:
Thank you for your instructive advice. Because I aim to condense the first two sections, I will delete all this sentences.
- line 109: please explain what the word “gross’’ is referring to here? Sounds like some medical jargon, which should be avoided.
Reply:
Thank you for your instructive advice. I will delete the word “gross” in this sentence, thank you.
- lines 223-225: “Furthermore, newly advanced nanocarrier-based combination therapy for GBMs has the additional advantages of facilitating sequential drug exposure, well confirmation of the synergistic drug ratio, and improving the localization of anticancer agents into the tumor site [57,58].’’ – please check the grammar and style of this sentence.
Reply:
Thank you for your instructive advice. I have revised this sentence as “The newly development of nanocarrier-based combination therapy for GBMs has additional advantages, including facilitation of sequential drug exposure, well confirmation of the synergistic drug ratio, and improved localization of anticancer agents into the tumor site [57,58].” (line 332-334)
- There is also a typo: line 59: GBS, should be GBM.
Reply:
Thank you for your correction, I will correct it. (line 59)
Reviewer 2 Report
The manuscript attempts to review recent nano-approaches for the treatment of glioblastoma multiforme (GBM). I found the manuscript is on a topic of relevance and general interest to the readers of the journal. The manuscript was well written and covered a wide range of information. However, the manuscript should only be considered for publication after the authors address all of the issues below.
First of all, the coherence between section 2 and sections 3/4 is not strong. Though the authors presented two main obstacles in treating GBM which were GBM stem cells and the transport through BBB, the authors did not really focus on how updated nano-approaches were used to overcome those issues. For example, there is no discussion on how nano-approaches have been used to tackle the GBM stem cells or how nanoparticles have been used to overcome the BBB? I strongly request the authors to revise sections 2, 3, and 4 to make it clear how nano-approaches have been used to solve the issues related to GBM treatment.
The authors only presented information on how nano-systems were designed and how the effects were but did not make any deep discussion on the advantages and disadvantages of each system. Please discuss more.
Some presented information is not relevant or inaccurate. For example, the authors mentioned that “…conjugation of polyethylene glycol (PEG) to the surface of a liposome 238 phospholipid bilayer can extend the half-life of liposomes in the circulation” (Lines 238-239), but it is not clear how it was contributed to improving GBM treatment. Or “poly (propylene glycol) (PPG)” (line 257) is not a biodegradable polyester.
The inclusion of section 4 does not add a lot of value to the manuscript. And in fact, hyperthermia and antitumor antibiotics are not emerging technologies for GBM. I recommend the authors to merge sections 3 and 4 into one.
Author Response
RE: Cancers-1037486
Nanotechnology and nanocarrier-based drug delivery as the potential therapeutic strategies for glioblastoma multiforme: an update
Dear Editor,
Thank you for your appreciated comments on our manuscript. We had the manuscript revised, all according to the reviewers’ and editor’s suggestions. We underline every change and highlight in red color on the revised manuscript. The replies for the reviewers’ criticisms are as followings. We hope this revised version can be acceptable.
Best regards,
Ming-Horng Tsai
Chief, Division of Neonatology and Pediatric Hematology/Oncology, Department of Pediatrics, Yunlin Chang Gung Memorial Hospital, Taiwan, R.O.C.
Comments from Reviewer No.2:
The manuscript attempts to review recent nano-approaches for the treatment of glioblastoma multiforme (GBM). I found the manuscript is on a topic of relevance and general interest to the readers of the journal. The manuscript was well written and covered a wide range of information. However, the manuscript should only be considered for publication after the authors address all of the issues below.
Reply:
Thank you for your appreciated comments and instructive advice. I appreciate the reviewer give me the chance to revise.
First of all, the coherence between section 2 and sections 3/4 is not strong. Though the authors presented two main obstacles in treating GBM which were GBM stem cells and the transport through BBB, the authors did not really focus on how updated nano-approaches were used to overcome those issues. For example, there is no discussion on how nano-approaches have been used to tackle the GBM stem cells or how nanoparticles have been used to overcome the BBB? I strongly request the authors to revise sections 2, 3, and 4 to make it clear how nano-approaches have been used to solve the issues related to GBM treatment.
Reply:
Thank you for your appreciated comments and instructive advice. I will add a section regarding how nano-approaches have been used to tackle the GBM stem cells or how nanoparticles have been used to overcome the BBB in page 7 and page 8, section 3.2 and section 3.3 (line 267-330 and relevant reference no. 89-114)
The authors only presented information on how nano-systems were designed and how the effects were but did not make any deep discussion on the advantages and disadvantages of each system. Please discuss more.
Reply:
Thank you for your instructive advice. I will discuss on the advantages and disadvantages of each system in the section 4, including line 342-345, line 363-366, line 371-374, line 378-380, line 396-401, line 435-441, line 443-447.
Some presented information is not relevant or inaccurate. For example, the authors mentioned that “…conjugation of polyethylene glycol (PEG) to the surface of a liposome 238 phospholipid bilayer can extend the half-life of liposomes in the circulation” (Lines 238-239), but it is not clear how it was contributed to improving GBM treatment. Or “poly (propylene glycol) (PPG)” (line 257) is not a biodegradable polyester.
Reply:
Thank you for your instructive advice. I will add “because PEG can help the nanoparticles escape from the capture of the RES [85]” in previous line 238 (now line 351 in the revised manuscript). Because I have mentioned in line 257-260 in the revised manuscript, I will not explain in full sentence in line 351.
Thank you for your correction. I have deleted PPG and used long-chain alkyl derivatives in the revised manuscript, thank you. (line 375).
The inclusion of section 4 does not add a lot of value to the manuscript. And in fact, hyperthermia and antitumor antibiotics are not emerging technologies for GBM. I recommend the authors to merge sections 3 and 4 into one.
Reply:
Thank you for your instructive advice. I have merged sections 3 and 4 into one, and as the new section 4 in the revised manuscript, thank you.
Reviewer 3 Report
Generally this was a well written and intersting article updating the use of nanocarriers for the treatment of GBM, a disease of high interest as generally new chemicals and technologies have failed to make a significant difference in treatment of decades. While the background is a little heavy most of the sections make a clear story as to the need and the point of this review paper which is done very well. In addition to just talking about the formulations the inclusion of the clinical study section is helpful in showing the translation of this research into becoming therapeutics for patients. While I have a few specific comments that follow, I have a few major comments on this manuscript that need to be addressed.
Major/General Comments:
While the background information shows a strong understanding about GBM and its current treatments it is too in depth and pulls away from the focus of the manuscript on nanocarrier treatments. I recommend removing the clinical features section all together.
In recent years there have been many articles and reviews about nanocarriers by N2B for treating brain disorders. I think this merits a section in this review paper discussing the feasibility or infeasibility of nanocarriers for GBM by this route of administration
Simply Summary section is too much like the abstract. This should be cut down into maybe 2 succinct sentences that really just expand on the title so a reader can decide if this is the article for them or not.
Language errors: There are many errors throughout the manuscript particularly with the use of "the" instead of "a" and "has" instead of "have"
- In the title "the " should be changed to "a" otherwising a conclusion about the success of nanocarriers against GBM is being made that is not yet substantiated
Specific comments
Abstract - remove nanoparticles conjugated to liposomes statement. I think this might be miswritten as I did not see this in the paper or it is at least not a significant portion to merit inclusion in the abstract.
Line 206, would include could also have interactions with other medications as well.
Line 264-266 ref 79,80 This section is set up as ongoing research is moving towards improved targeting of the polymeric micelle drug delivery systems and this is listed as a conclusion of this study with 2 different pluronics in line 268. It is unclear how improved targeting is being achieved, with just the use of Pluronics? Discussion of this study should either be set up differently to highlight the conclusions of these papers or more detail into how they are specifically targeting should be provided.
3.4 metal particles There have been several concerns over the years about clearance and accumulation of these particles in the body. A description of their potential limitations or whether or not they have been proven safe would be a nice addition to this section.
Author Response
RE: Cancers-1037486
Nanotechnology and nanocarrier-based drug delivery as the potential therapeutic strategies for glioblastoma multiforme: an update
Dear Editor,
Thank you for your appreciated comments on our manuscript. We had the manuscript revised, all according to the reviewers’ and editor’s suggestions. We underline every change and highlight in red color on the revised manuscript. The replies for the reviewers’ criticisms are as followings. We hope this revised version can be acceptable.
Best regards,
Ming-Horng Tsai
Chief, Division of Neonatology and Pediatric Hematology/Oncology, Department of Pediatrics, Yunlin Chang Gung Memorial Hospital, Taiwan, R.O.C.
Comments from Reviewer No.3:
Generally this was a well written and interesting article updating the use of nanocarriers for the treatment of GBM, a disease of high interest as generally new chemicals and technologies have failed to make a significant difference in treatment of decades. While the background is a little heavy most of the sections make a clear story as to the need and the point of this review paper which is done very well. In addition to just talking about the formulations the inclusion of the clinical study section is helpful in showing the translation of this research into becoming therapeutics for patients. While I have a few specific comments that follow, I have a few major comments on this manuscript that need to be addressed.
Reply:
Thank you for your appreciated comments and instructive advice.
Major/General Comments:
While the background information shows a strong understanding about GBM and its current treatments it is too in depth and pulls away from the focus of the manuscript on nanocarrier treatments. I recommend removing the clinical features section all together.
Reply:
Thank you for your appreciated comments and instructive advice. I will try to condense the clinical features and current treatment section of GBM as another reviewer also has suggested. However, I beg not to remove all this section, because I found most of the review articles published in “Cancers” and most of the review articles regarding nanotechnology or nanocarriers of GBM have the clinical features as the introduction
In recent years there have been many articles and reviews about nanocarriers by N2B for treating brain disorders. I think this merits a section in this review paper discussing the feasibility or infeasibility of nanocarriers for GBM by this route of administration
Reply:
Thank you for your instructive advice. I will add a section to discuss the feasibility and infeasibility of nanocarriers for GBS by this route of administration (page 7 and page 8, line 270 to line 290).
Simply Summary section is too much like the abstract. This should be cut down into maybe 2 succinct sentences that really just expand on the title so a reader can decide if this is the article for them or not.
Reply:
Thank you for your instructive advice. The simple summary is the unique format of the review article in the journal “Cancers”. The “Cancers” journal requests a mandatory “Simple summary” in front of the abstract, with 100-150 words to summarize the article. Therefore, it is very difficult to cut down into 2 succinct sentences. I have tried to make the simple summary different from the abstract, but I think simple summary is basically similar to abstract to let a reader can decide if this is the article for them or not.
Language errors: There are many errors throughout the manuscript particularly with the use of "the" instead of "a" and "has" instead of "have"
Reply:
Thank you for your instructive advice. I will revise all the language errors and use the correct one in the revised manuscript accordingly, thank you. If I still make a mistake in language errors in the revised manuscript, the language editing of the journal “Cancers” in the final version will be done.
In the title "the " should be changed to "a" otherwising a conclusion about the success of nanocarriers against GBM is being made that is not yet substantiated
Reply:
Thank you for your instructive advice. I will revise the title to be “Nanotechnology and nanocarrier-based drug delivery as a potential therapeutic strategy for glioblastoma multiforme: an update”.
Specific comments
Abstract - remove nanoparticles conjugated to liposomes statement. I think this might be miswritten as I did not see this in the paper or it is at least not a significant portion to merit inclusion in the abstract.
Reply:
Thank you for your instructive advice. I will remove nanoparticles conjugated to liposomes statement, and revise as Current studies using nanoparticles or nanocarrier-based drug delivery system for treatment of GBMs in clinical trials, as well as the advantages and limitations are also reviewed. (line 43-44). thank you.
Line 206, would include could also have interactions with other medications as well.
Reply:
Thank you for your instructive advice. I have added “For example, statins can reduce the efflux activity of Pgp and BCRP by increasing NO synthesis, which have been documented in statins plus doxorubicin-loaded nanoparticles to be efficient vehicle to cross the BBB [68].” in the revised manuscript (line 197-199).
Line 264-266 ref 79,80 This section is set up as ongoing research is moving towards improved targeting of the polymeric micelle drug delivery systems and this is listed as a conclusion of this study with 2 different pluronics in line 268. It is unclear how improved targeting is being achieved, with just the use of Pluronics? Discussion of this study should either be set up differently to highlight the conclusions of these papers or more detail into how they are specifically targeting should be provided.
Reply:
Thank you for your question. Actually the targeting ability of these two pluronics come from the presence of 17-AAG, and nanomaterials-based drug delivery carriers can overcome the drawbacks of use 17-AAG only. I will revise this sentence as “17-AAG is a potent inhibitor of heat shock protein 90 (Hsp90) and can cause destabilization of Hsp90 related client proteins in cancer cells” to make it clear in the revised manuscript (line 384-385), thank you.
3.4 metal particles There have been several concerns over the years about clearance and accumulation of these particles in the body. A description of their potential limitations or whether or not they have been proven safe would be a nice addition to this section.
Reply:
Thank you for your instructive advice. I will add the description of the potential limitations and clearance and accumulation of metal particles in the section of metal particles. (line 435-line 441)
Round 2
Reviewer 1 Report
The manuscript has been significantly improved.
Reviewer 2 Report
The authors well addressed the issues of the manuscript. The manuscript should be accepted for publication.